# Pain persists in mice lacking both Substance P and CGRPα signaling

Donald Iain MacDonald[1], Monessha Jayabalan[1], Jonathan T Seaman[1], Rakshita Balaji[1], Alec R Nickolls[1], Alexander Theodore Chesler[1,2]*

[1]National Center for Complementary and Integrative Health, National Institutes of Health, Bethesda, United States; [2]National Institute of Neurological Disorders and Stroke, National Institutes of Health, Bethesda, United States

## eLife Assessment

This report used a new double knockout mouse model to investigate the role of two neuropeptides, substance P and CGRPa, in pain signaling. There is **convincing** evidence that double knockout of these two molecules, both of which have historically been associated with pain, does not affect nociception or acute pain behaviors in males and females. This finding is **fundamental**, as it challenges the hypothesis that these peptides are essential for pain transmission, even when targeted together. This paper will be of interest to those interested in the neurobiology of pain and/or neuropeptide function.

*For correspondence:
alexander.chesler@nih.gov

**Abstract** The neuropeptides Substance P and CGRPα have long been thought important for pain sensation. Both peptides and their receptors are expressed at high levels in pain-responsive neurons from the periphery to the brain making them attractive therapeutic targets. However, drugs targeting these pathways individually did not relieve pain in clinical trials. Since Substance P and CGRPα are extensively co-expressed, we hypothesized that their simultaneous inhibition would be required for effective analgesia. We therefore generated *Tac1* and *Calca* double knockout (DKO) mice and assessed their behavior using a wide range of pain-relevant assays. As expected, Substance P and CGRPα peptides were undetectable throughout the nervous system of DKO mice. To our surprise, these animals displayed largely intact responses to mechanical, thermal, chemical, and visceral pain stimuli, as well as itch. Moreover, chronic inflammatory pain and neurogenic inflammation were unaffected by loss of the two peptides. Finally, neuropathic pain evoked by nerve injury or chemotherapy treatment was also preserved in peptide-deficient mice. Thus, our results demonstrate that even in combination, Substance P and CGRPα are not required for the transmission of acute and chronic pain.

## Introduction

Over a hundred neuropeptides are expressed by mammalian neurons (*Russo, 2017*). Neuropeptides are released through the secretory pathway and activate G-protein coupled receptors to control neuronal excitability and synaptic strength (*Pagani et al., 2019*). They can also influence the function of immune cells in peripheral tissues (*Chiu et al., 2012*). Drugs acting on neuropeptides or their receptors are now widely used in the clinic, including therapeutics for obesity and migraine (*Drucker, 2022*; *Ogunlaja and Goadsby, 2022*). Which peptides can be targeted successfully and for what indications thus remain key questions for neuroscience and drug development.

Among the most challenging diseases to treat is chronic pain. With over 20% of the population suffering from chronic pain, we urgently need to find new analgesic targets (*Nahin et al., 2023*).

The two neuropeptides most strongly implicated in chronic pain are Substance P and CGRPα (*De Matteis et al., 2020*; *Paige et al., 2022*; *Yaksh et al., 1980*; *Zieglgänsberger, 2019*). Substance P is an 11-amino acid peptide first discovered as a tissue extract with contractile activity (*V Euler and Gaddum, 1931*). CGRPα is a 37-amino acid peptide and potent vasodilator identified as the alternatively spliced product of the calcitonin gene (*Amara et al., 1982*). Both Substance P and CGRPα are highly expressed in pain-responsive neurons throughout the nervous system. In the periphery, strong activation of nociceptor sensory neurons is reported to cause peptide secretion and neurogenic inflammation (*Chiu et al., 2012*). Similarly, release of Substance P and CGRPα in the spinal cord and brainstem may alter pain transmission in ascending pain pathways (*Huang et al., 2019*; *Latremoliere and Woolf, 2009*; *Löken et al., 2021*). In the brain, these molecules are prominently enriched in areas known to be involved in pain, including the periaqueductal grey, parabrachial nucleus and amygdala (*Barik et al., 2018*; *Lein et al., 2007*; *Palmiter, 2018*). Thus, decades of research implicate Substance P and CGRPα as critical drivers of pain, including modulating tissue inflammation, sensory hypersensitivity, pain chronification and unpleasantness (*Chiu et al., 2012*; *De Matteis et al., 2020*; *Zieglgänsberger, 2019*).

However, selective antagonists of the Substance P receptor NKR1 failed to relieve chronic pain in human clinical trials (*Hill, 2000*). Although CGRP monoclonal antibodies and receptor blockers have proven effective for subsets of migraine patients, their usefulness for other types of pain in humans is unclear (*De Matteis et al., 2020*; *Jin et al., 2018*). In line with this, knockout mice deficient in Substance P, CGRPα or their receptors have been reported to display some pain deficits, but the analgesic effects are neither large nor consistent between studies (*Cao et al., 1998*; *De Felipe et al., 1998*; *Guo et al., 2012*; *Salmon et al., 2001*; *Salmon et al., 1999*; *Zimmer et al., 1998*).

By contrast, ablating or inhibiting peptidergic nociceptors causes profound decreases in heat sensitivity, inflammatory heat hyperalgesia, and some forms of mechanical allodynia (*Cowie et al., 2018*; *McCoy et al., 2013*). Mice with silenced Substance P or CGRPα-expressing neurons in brain areas such as the parabrachial nucleus also show impaired pain behavior (*Barik et al., 2018*; *Han et al., 2015*; *Kang et al., 2022*; *Sun et al., 2020*). In the periphery, Substance P and CGRPα are expressed by largely the same nociceptors, in mice and in humans (*Nguyen et al., 2021*; *Sharma et al., 2020*). The two peptides are also co-expressed in the brain (*Pauli et al., 2022*; *Zeisel et al., 2018*). Redundancy is built into pain pathways because of their importance for survival and may explain why attenuating the signaling of a single peptide does not recapitulate the analgesic effect of silencing the cell secreting it (*Vandewauw et al., 2018*).

Given the well-described roles for peptidergic neurons in pain and the widespread co-expression of Substance P and CGRPα, we reasoned that removal of both peptides should reveal roles for peptidergic signaling that might be masked by redundancy. We therefore generated and characterized *Tac1::Calca* double knockout (DKO) mice lacking both Substance P and CGRPα peptides. We predicted these DKO animals might recapitulate the striking absence of inflammatory pain observed in naked mole rats that release glutamate but not Substance P or CGRPα from nociceptors (*Park et al., 2008*; *Park et al., 2003*). Here, we present a thorough evaluation of the impact of dual peptide deletion on acute and chronic pain behavior in laboratory mice.

## Results

### Double knockout mice completely lack Substance P and CGRPα signaling

To investigate the roles of Substance P and CGRPα in pain processing, we generated Tac1::Calca Double Knockout (DKO) mice with constitutive deletion of the two peptide precursor genes. A homozygous Tac1-RFP knockin-knockout line was used to remove the *Tac1* gene encoding preprotachykinin, the precursor for Substance P and Neurokinin A (*Wu et al., 2018*). As expected, no detectable Substance P immunostaining was observed in pain-relevant areas including the dorsal root ganglion, dorsal horn of the spinal cord, parabrachial nucleus and periaqueductal gray (*Figure 1A*). To eliminate CGRPα signaling, we used a Calca-Cre mouse, which at homozygosity is a knockout for the *Calca* gene encoding the Calcitonin and CGRPα peptides (*Carter et al., 2013*; *Chen et al., 2018*). We were unable to detect CGRP immunoreactivity in DRG, spinal cord and the amygdala (*Figure 1B*) validating the DKO approach.

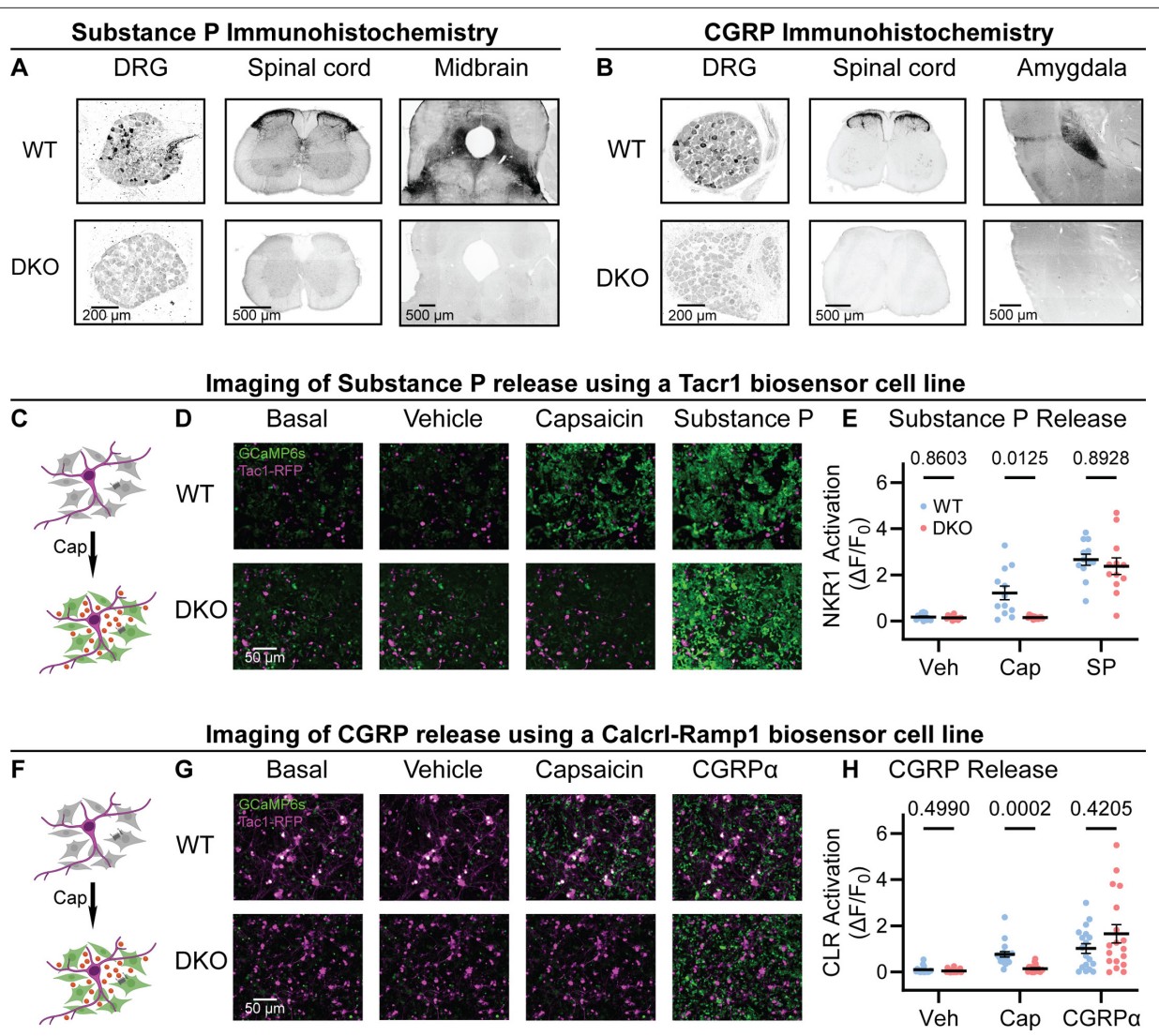

**Figure 1.** Tac1::Calca DKO mice lack Substance P and CGRPα peptides throughout the nervous system. (**A**) Confocal images showing Substance P immunostaining in the dorsal root ganglion (DRG), spinal cord dorsal horn and midbrain of a WT mouse (*top*). No staining is detectable in the DKO (*bottom*). (**B**) Confocal images showing CGRP immunostaining in the DRG, spinal cord dorsal horn, and amygdala of a WT mouse (*top*). The DKO mice show no obvious staining (*bottom*). Images in both (A) and (B) are representative of staining performed on tissue from 1 M and 1 F animal per genotype. (**C**) Schematic showing co-culture of Tac1-RFP-labeled DRG neurons and Substance P-sniffer HEK293 cells expressing NKR1, GCaMP6s, and Gαo. Capsaicin-evoked secretion of Substance P from DRG activates the NKR1 receptor in neighboring HEK leading to calcium release from stores and GCaMP6s-mediated fluorescence increase. (**D**) Fluorescence images showing Tac1-RFP-labeled DRG neurons (*magenta*) and Substance P-sniffer cells (*green*). Capsaicin causes an increase in Substance P-sniffer GCaMP6s fluorescence when cultured with DRGs from WT, but not DKO, mice. Application of exogenous Substance P (10 nM) activates Substance P-sniffers in both conditions. (**E**) Quantification of fluorescence change in Substance P-sniffer cells in response to vehicle, capsaicin, and Substance P stimulation in WT and DKO mice. n=12 wells from 2 mice (1 M, 1 F) for WT, and n=12 wells from 2 mice (1 M, 1 F) for DKO. For (**E**), means were compared using repeated measures 2-way ANOVA, followed by post-hoc Sidak's test. Error bars denote standard error of the mean. (**F**) Schematic showing co-culture of Tac1-RFP-labeled DRG neurons and Substance P-sniffer HEK293 cells expressing Calcrl, Ramp1, GCaMP6s, and Gαo. Capsaicin-evoked secretion of CGRPα from DRG activates the CGRP receptor complex in neighboring HEK leading to calcium release from stores and GCaMP6s-mediated fluorescence increase. (**G**) Fluorescence images showing Tac1-RFP-labeled DRG neurons (*magenta*) and Substance P-sniffer cells (*green*). Capsaicin causes an increase in CGRP-sniffer GCaMP6s fluorescence when cultured with DRGs from WT, but not DKO, mice. Application of exogenous CGRPα at a saturating concentration (100–1000 nM) activates CGRP-sniffers in both conditions. (**H**) Quantification of fluorescence change in CGRP-sniffer cells in response to vehicle, capsaicin, and CGRPα stimulation in WT and DKO mice. n=18 wells from 3 mice (2 M, 1 F) for WT, and n=18 wells from 3 mice (2 M, 1 F) for DKO. For (**H**), means were compared using repeated measures two-way ANOVA, followed by post-hoc Sidak's test. Error bars denote standard error of the mean.

The online version of this article includes the following figure supplement(s) for figure 1:

**Figure supplement 1.** Substance P-sniffer cells selectively respond to Substance P.

A clear prediction from the loss of immunostaining is that sensory neuron activity should no longer activate the neuropeptide receptors. We began by developing a cell-based assay to monitor signaling through the Substance P receptor (NKR1) by generating HEK293 cells stably expressing NKR1, GCaMP6s and Gα15 (so-called 'Substance P-sniffer' cells; *Figure 1C*). These cells responded to low nanomolar concentrations of Substance P (EC$_{50}$=21 nM, *Figure 1—figure supplement 1A*), but were insensitive to CGRPα. We reasoned that in co-cultures with DRG neurons, application of the potent TRPV1 agonist capsaicin (10 µM) would stimulate nociceptors to release Substance P and thereby activate neighboring Substance P-sniffer cells. Control DRGs treated with capsaicin evoked robust GCaMP responses in surrounding Substance P-sniffer cells (*Figure 1D–E*). In contrast, capsaicin stimulated DKO-DRG neurons induced no Substance P-sniffer cell responses (*Figure 1D–E*).

We also developed a cell-based approach to monitor CGRP release from nociceptors. The CGRP receptor consists of two co-receptors encoded by the genes *Calcrl* and *Ramp1*. We therefore engineered a stable inducible HEK293 sniffer cell line that expressed Calcrl, Ramp1, GCaMP6s and Gα15. These cells responded to low nanomolar concentrations of CGRPα (EC$_{50}$=27 nM, *Figure 1—figure supplement 1B*), but were insensitive to Substance P. When we cultured WT DRG neurons with the

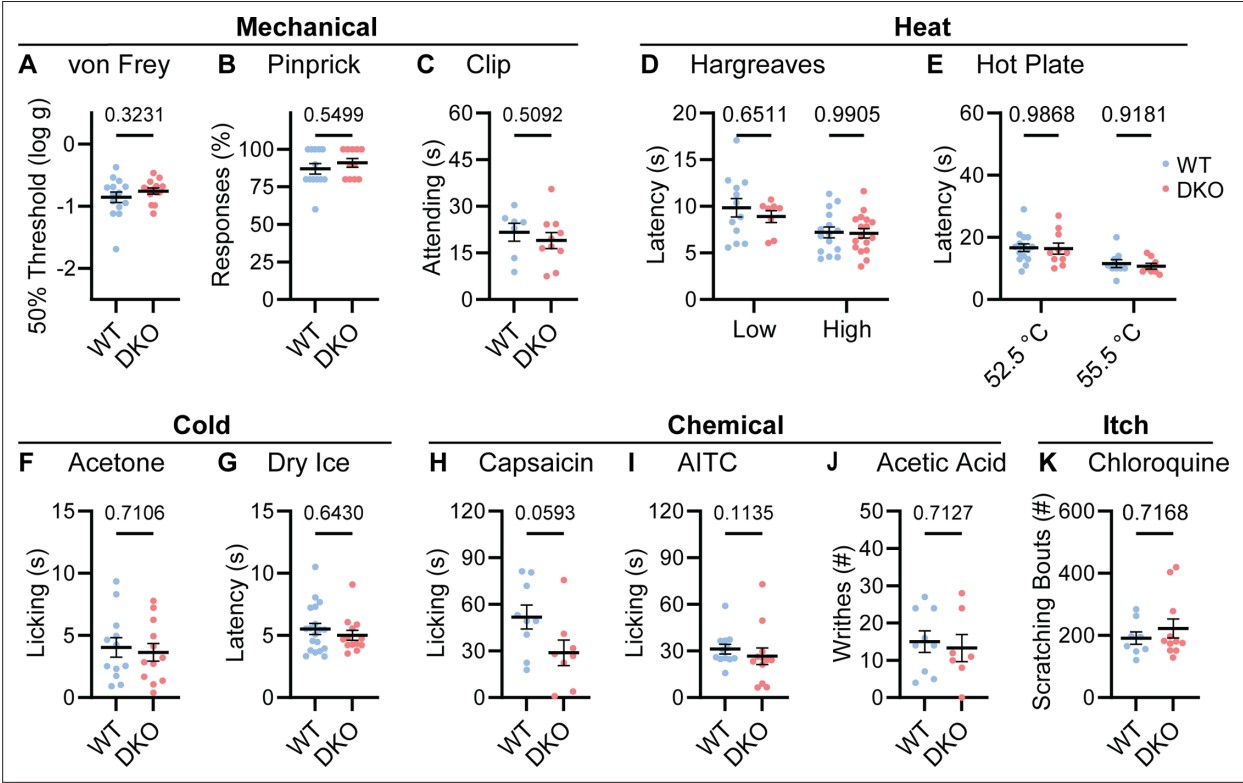

**Figure 2.** Tac1::Calca DKO mice respond to acute painful and itching stimuli. (**A**) 50% withdrawal threshold for von Frey punctate stimulation (log g). n=14 (7M, 7 F) for WT & n=14 (7M, 7 F) for DKO. (**B**) Percentage response to noxious pinprick stimulation. n=13 (5M, 8 F) for WT and n=11 (4M, 7 F) for DKO. (**C**) Time spent attending to alligator clip applied to paw for 60 s. n=7 (3M, 4 F) for WT and n=10 (6M, 4 F) for DKO. (**D**) Latency to withdraw to low and high radiant heat. For low setting, n=12 (6M, 6 F) for WT and n=8 (4M, 4 F) for DKO. For high setting, n=15 (8M, 7 F) for WT and n=17 (9M, 8 F) for DKO. (**E**) Latency to lick hindpaw following exposure to hot plate at two different temperatures. For 52.5 °C, n=15 (7M, 8 F) for WT and n=10 (6M, 4 F) for DKO. For 55.5 °C, n=9 (6M, 3 F) for WT and n=8 (5M, 3 F) for DKO. (**F**) Time spent licking in 60 s immediately following acetone application to paw. n=12 (5M, 7 F) for WT and n=12 (5M, 7 F) for DKO. (**D**) Latency to respond to dry ice. n=19 (8M, 11 F) for WT and n=13 (5M, 8 F) for DKO. (**H**) Time spent licking in 5 min after capsaicin injection to paw. n=9 (4M, 5 F) for WT and n=8 (4M, 4 F) for DKO. (**I**) Time spent licking in 5 min after 1% AITC injection to paw. n=12 (6M, 6 F) for WT and n=12 (6M, 6 F) for DKO. (**J**) Number of writhes in 15 min starting 5 min after intraperitoneal injection of 0.6% acetic acid. n=9 (5M, 4 F) for WT and n=7 (5M, 2 F) for DKO. (**K**) Scratching bouts in 15 min evoked by chloroquine injection into the nape of the neck. n=8 for WT and n=11 for DKO. For (**A, C, F, H, and J**) means were compared using unpaired *t*-test, for (**B, G, I, and K**) a Mann-Whitney U test was used, and for (**D-E**) a two-Way ANOVA followed by post-hoc Sidak's test was used. Error bars denote standard error of the mean.

The online version of this article includes the following figure supplement(s) for figure 2:

**Figure supplement 1.** Tac1::Calca DKO mice develop LiCl-induced conditioned taste aversion.

**Figure supplement 2.** Capsaicin evokes Fos activity in the dorsal horn of Tac1::Calca DKO mice.

CGRP-sniffer cells, we observed strong increases in calcium consistent with CGRP receptor activity in response to capsaicin application (*Figure 1F–H*). However, DKO neurons evoked markedly reduced activity in the adjacent sniffer cells (*Figure 1D–E*). Thus, *Calca* deletion largely abolished endogenous CGRPα signaling. In combination, these experiments demonstrate that deletion of *Tac1* and *Calca* completely blocks the endogenous activation of Substance P and CGRP receptors by nociceptor neurons.

## Loss of Substance P and CGRPα does not affect acute pain or itch

To understand whether neuropeptide signaling is required for pain sensation, we performed a comprehensive battery of somatosensory behavior tests on DKO and wild-type control mice (WT). Notably, regardless of the type of mechanical or thermal stimulus, groups of animals showed no difference in their behavior (*Figure 2*). Specifically, responses to mechanical stimulation by von Frey hairs, pinpricks or an alligator clip were quantitatively indistinguishable (*Figure 2A–C*). In addition, withdrawal responses to radiant heat stimulation (Hargreaves', *Figure 2D*), and to two noxious Hot Plate temperatures were unaffected in DKO mice (*Figure 2E*). Responses to cold evoked by acetone and dry ice were also the same between groups (*Figure 2F–G*). Intraplantar injection of the Trpv1 agonist capsaicin evoked coping-like licking responses in both genotypes; however, responses were variable and although decreased licking was observed in some DKO mice, differences did not reach statistical significance (*Figure 2H*). Therefore, to explore this potential difference using an independent measure, we quantified capsaicin-evoked Fos staining as a correlate of dorsal horn network activation. No differences in dorsal horn Fos expression were observed (*Figure 2—figure supplement 1A–B*). Furthermore, intraplantar injection of a different algogen (the Trpa1 agonist allyl isothiocyanate, AITC) evoked similar licking behavior to capsaicin that was unaffected by peptide deletion (*Figure 2I*). Together these data suggest that loss of the two neuropeptides does not alter acute withdrawal or coping-like responses to multiple modalities of damaging stimuli applied to the paw.

We investigated whether dual peptide deletion might attenuate visceral pain. First, we dosed mice with acetic acid via intraperitoneal injection. This evoked comparable writhing behavior in both WT and DKO mice (*Figure 2J*). Next, because NKR1 antagonists are effective anti-emetics, we wondered whether the two peptides contribute to nausea-like behavior in mice (*Warr et al., 2005*). In many species, lithium chloride induces gastrointestinal malaise and when paired with a normally attractive tastant produces a marked conditioned taste aversion (*Garcia et al., 1955*). Notably, pairing saccharin with LiCl injection elicited a strong and indistinguishable conditioned taste aversion in both WT and DKO mice (*Figure 2—figure supplement 2A*).

NKR1 antagonists have also been proposed to treat itch (*Pojawa-Gołąb et al., 2019*). Therefore, we examined whether peptide deletion abrogated itching. DKO mice scratched robustly following intradermal chloroquine injection, a response indistinguishable from WT animals (*Figure 2K*). Thus, our data demonstrate that Substance P and CGRPα contribute little to defensive behaviors evoked by diverse sensory stimuli.

## Inflammatory pain and neurogenic inflammation are preserved in Substance P and CGRPα-deficient mice

Chronic inflammatory pain results in long-lasting changes in nociceptor function and the downstream pathways they engage. This plasticity is widely thought to involve neuropeptides (*Zieglgänsberger, 2019*). We therefore tested whether loss of Substance P and CGRPα impacted the development of heat hypersensitivity following treatment with Complete Freund's Adjuvant (CFA). Surprisingly, WT and DKO mice with intraplantar injection of CFA developed strong heat hypersensitivity (*Figure 3A*). Mechanical hypersensitivity was also unaffected by peptide loss (*Figure 3B*). CFA injection causes a long-term inflammatory immune response, and therefore, we also tested Prostaglandin E2 (PGE2), which can acutely sensitize nociceptors. PGE2 injection elicited short-lasting heat and mechanical hypersensitivity to a similar extent in both WT and DKO mice (*Figure 3C–D*). Together, these data indicate that Substance P and CGRPα are not required for the behavioral sensitization associated with both acute and chronic inflammation.

Release of vasoactive peptides including Substance P and CGRPα from nociceptor terminals has been proposed to drive neurogenic inflammation, encompassing edema and extravasation (*Chiu et al., 2012*). Unexpectedly, we noticed that injection of the inflammatory mediators CFA and PGE2,

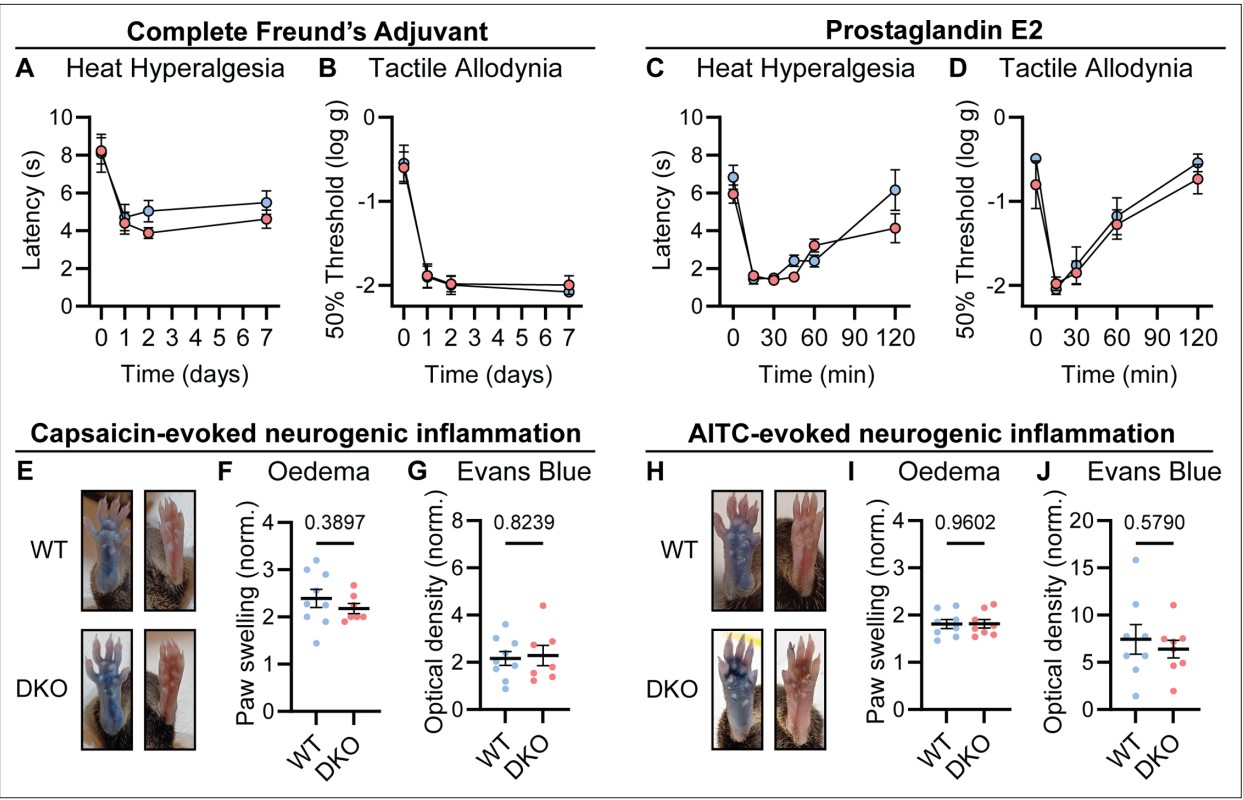

**Figure 3.** Tac1::Calca DKO mice display inflammatory pain and neurogenic inflammation. (**A**) Time course of the change in the Hargreaves' radiant heat withdrawal latencies of the hindpaw of WT and DKO mice following intraplantar injection of Complete Freund's Adjuvant (CFA). n=10 (5M, 5 F) for WT and n=9 (4M, 5 F) for DKO. (**B**) Time course of the change in von Frey 50% withdrawal thresholds (log g) on the hindpaw of WT and DKO mice after CFA. n=6 (3M, 3 F) for WT and n=6 (3M, 3 F) for DKO. (**C**) Time course of the change in the Hargreaves' radiant heat withdrawal latencies of the hindpaw of WT and DKO mice following intraplantar injection of Prostaglandin E2 (PGE2). n=10 (8M, 2 F) for WT and n=10 (7M, 3 F) for DKO. Post-hoc tests show only the 120 min time-point shows a significant difference (p=0.025). (**D**) Time course of the change in von Frey 50% withdrawal thresholds (log g) on the hindpaw of WT and DKO mice after PGE2. n=6 (3M, 3 F) for WT and n=6 (3M, 3 F) for DKO. (**E**) Images showing both WT and DKO mice show paw swelling and plasma extravasation following capsaicin injection (left) compared to uninjected paw (right). (**F**) Capsaicin-induced edema, with injected paw swelling measured by volume and normalized to the uninjected paw, in WT and DKO mice. n=9 for WT (5 M, 4 F) and n=7 for DKO (4 M, 3 F). (**G**) Optical density of Evans blue dye extracted from the capsaicin-injected paw, normalized to uninjected paw, in WT and DKO mice. n=9 for WT (5 M, 4 F) and n=7 for DKO (4 M, 3 F). (**H**) Images showing both WT and DKO mice show paw swelling and plasma extravasation following AITC injection (left) compared to uninjected paw (right). (**I**) AITC-induced edema, with injected paw swelling measured by volume and normalized to the uninjected paw, in WT and DKO mice. n=8 for WT (4 M, 4 F) and n=8 for DKO (4 M, 4 F). (**J**) Optical density of Evans blue dye extracted from AITC-injected paw, normalized to uninjected paw, in WT and DKO mice. n=8 for WT (4 M, 4 F) and n=8 for DKO (4 M, 4 F). For (**A-D**), means were compared using two-way ANOVA followed by post-hoc Sidak's test, and for (**F-G**) and (**I-J**), an unpaired t-test was used. Error bars denote standard error of the mean.

as well as the algogens AITC and capsaicin, provoked swelling of the hindpaw of DKO mice to a similar extent as WT mice. To examine the development of neurogenic inflammation more rigorously, we measured paw volumes after capsaicin injection, and observed no appreciable differences in capsaicin-induced swelling between WT and DKO mice. (*Figure 3E–F*). In addition, using the Evans blue dye method, we found that plasma extravasation was intact in DKO mice (*Figure 3G*). Similar results were observed following AITC injection (*Figure 3H–J*). Therefore, Substance P and CGRPα are dispensable for some forms of neurogenic inflammation.

## Neuropathic pain is unaffected by Substance P and CGRPα deletion

Lastly, we investigated how dual peptide deletion affected the development of allodynia symptoms, where innocuous stimuli are perceived as painful, in two mouse models of neuropathic pain (*Jensen and Finnerup, 2014*). First, we performed a sciatic spared nerve injury on DKO animals and found that the static mechanical allodynia evoked by von Frey filaments developed to the same degree as in WT mice, persisting for 3 weeks (*Figure 4A*). Brush-evoked Fos activity in the superficial laminae of the dorsal horn of SNI-treated mice – a correlate of dynamic allodynia – was also comparable between

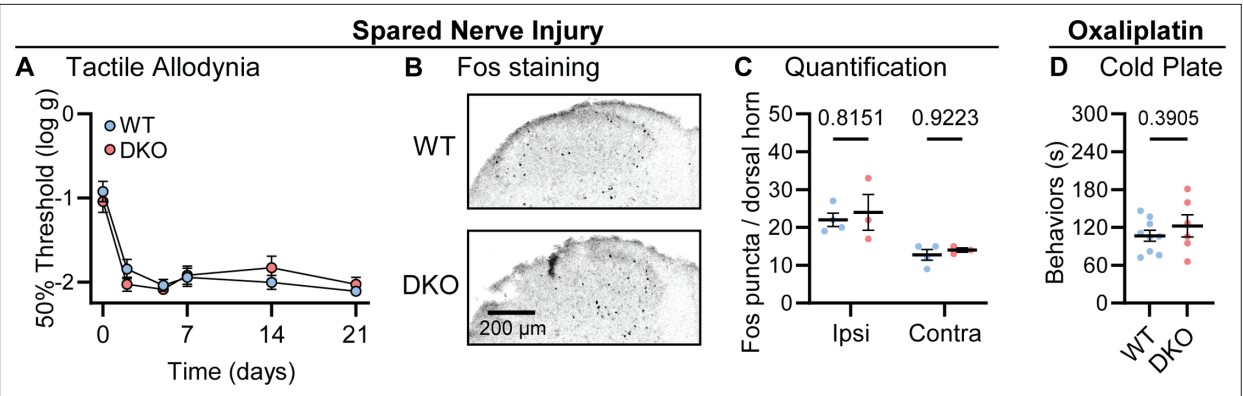

**Figure 4.** Tac1::Calca DKO mice develop neuropathic pain associated with nerve injury and the chemotherapeutic drug oxaliplatin. (**A**) Time course of the change in the von Frey 50% withdrawal threshold (log g) of the hindpaw of WT and DKO mice after sciatic spared nerve injury. n=8 (4M, 4 F) for WT and n=8 (4M, 4 F) for DKO. (**B**) Example confocal images showing Fos staining in the dorsal horn of WT and DKO mice following 30 min of brushing of the lateral part of the plantar surface of the hindpaw ipsilateral to the spared nerve injured. Fos puncta are visible in the ipsilateral dorsal horn of both WT and DKO cases. (**C**) Quantification of the mean number of Fos puncta in the ipsilateral and contralateral dorsal horn of WT and DKO mice. For each mouse, the number of Fos puncta was counted in five sections and then averaged so that n is the number of mice. n=4 for WT and n=3 for DKO. (**D**) Quantification of the time WT and DKO mice pre-treated with oxaliplatin (40 µg / 40 µl intraplantar) spent exhibiting pain-like behaviors in 5 min of exposure to a cold plate held at –10 °C. n=9 (5M, 4 F) for WT and n=6 (3M, 3 F) for DKO. Means were compared for (**A**) with a two-Way ANOVA followed by post-hoc Sidak's test, and for (**D**) with an unpaired t test. Error bars denote standard error of the mean.

genotypes (*Figure 4B–C*). Second, we treated mice with the chemotherapeutic drug oxaliplatin which evokes extreme cold allodynia that is known to depend, in part, on CGRPα-positive silent cold-sensing neurons (*MacDonald et al., 2021*). One day after intraplantar oxaliplatin treatment, both DKO and WT animals displayed pronounced pain-like behaviors when placed on a Cold Plate held at 10 °C (*Figure 4D*). Substance P and CGRPα are therefore not essential for mechanical or cold allodynia.

## Discussion

The neuropeptides Substance P and CGRPα have been proposed to play diverse but largely overlapping roles in acute, inflammatory, and neuropathic pain. However, when the levels of peptide signaling have been experimentally manipulated, effects have often been small and variable between studies, possibly due to redundancy (*Hohmann et al., 2004*). We therefore generated DKO mice lacking both Substance P and CGRPα signaling. Remarkably, both peptides were dispensable for pain across a wide range of assays.

It is difficult to reconcile our findings with the fact that Substance P and CGRPα are highly expressed throughout ascending pain pathways and are often found together in the same cells. (*Pauli et al., 2022*; *Sharma et al., 2020*; *Zeisel et al., 2018*). Studies of *Tac1* and *Calca* single KO mice reported significant impairments in acute pain, including heat sensitivity, visceral pain and the formalin test (*Cao et al., 1998*; *Salmon et al., 2001*; *Salmon et al., 1999*; *Zimmer et al., 1998*). Notably, these deficits were modest, and in fact inconsistent between studies (*Woolf et al., 1998*; *Zajdel et al., 2021*). Nonetheless, ablating either peripheral or central neurons expressing Substance P and CGRPα produces profound analgesia (*Barik et al., 2018*; *Cowie et al., 2018*; *Han et al., 2015*; *McCoy et al., 2013*). Importantly, these neurons are all glutamatergic and are thought to co-release neuropeptides to modulate synaptic transmission and neuronal firing (*Pagani et al., 2019*). Our work thus substantiates previous findings that established primary afferent-derived glutamate as the critical transmitter for most pain sensations (*Lagerström et al., 2010*; *Lagerström et al., 2011*; *Liu et al., 2010*; *Rogoz et al., 2012*; *Scherrer et al., 2010*), and suggests that the two peptides play at most a minor role.

An oft proposed caveat is that constitutive deletion of genes could be compensated for by upregulating the expression of functionally similar molecules, but with surprisingly little evidence (*El-Brolosy and Stainier, 2017*). Alternatively, neuropeptides may have opposing effects in different parts of the circuitry meaning global loss of the gene may produce a net effect of no change in a particular behavior. Indeed, classical studies show Substance P infusion into the spinal cord elicits pain, but paradoxically in the brain is analgesic (*Hylden and Wilcox, 1981*; *Malick and Goldstein, 1978*). Despite

this, knockout of a gene remains the strongest test of whether the molecule it encodes is essential for a biological phenomenon (*Caterina et al., 2000*; *Chesler et al., 2016*; *Mishra and Hoon, 2013*), and our results clearly demonstrate that Substance P and CGRPα are not required for pain transmission.

The striking and highly-conserved pattern of expression of these two peptides in pain pathways, particularly in nociceptors, raises the question of why these neurons evolved to release them. Rather than directly acting as pain transmitters in the CNS, accumulating evidence indicates that the secretion of these neuropeptides from nociceptor peripheral terminals modulates immune cells and the vasculature in diverse tissues (*Chiu et al., 2013*; *Chiu et al., 2012*; *Cohen et al., 2019*; *Lai et al., 2020*; *Perner et al., 2020*; *Pinho-Ribeiro et al., 2016*; *Yang et al., 2022*). We focused on pain transmission, but it is clear our DKO mice will be useful reagents for exploring the crosstalk between nociceptors and other body systems. For example, the development of effective migraine therapeutics targeting CGRPα or its receptor confirm the important role this peptide plays in headache (*De Matteis et al., 2020*; *Tso and Goadsby, 2017*), and new efferent functions for both CGRPα and Substance P are regularly being uncovered (*Brain, 1997*; *Caceres et al., 2009*; *Perner et al., 2020*; *Pinho-Ribeiro et al., 2016*; *Wilhelms et al., 2018*; *Yang et al., 2022*).

Beyond Substance P and CGRPα, pain-responsive neurons express a rich repertoire of potential signaling molecules, including other neuropeptides. Emerging approaches to image and manipulate these molecules (*Girven et al., 2022*; *Kim et al., 2024*), as well as advances in quantitating pain behaviors (*Bohic et al., 2023*; *MacDonald and Chesler, 2023*), may ultimately reveal the fundamental roles of neuropeptides in generating our experience of pain.

## Materials and methods

### Animals
Animal care and experimental procedures were performed in accordance with a protocol approved by the National Institute for Neurological Diseases and Stroke (NINDS) Animal Care and Use Committee. DKO mice were generated by crossing B6.Cg-Calcatm1.1(cre/EGFP)Rpa/J (Jax #033168) with Tac1-tagRFP-2A-TVA (*Carter et al., 2013*; *Wu et al., 2018*). Control WT animals were C57BL/6 J mice from The Jackson Laboratory (Jax #000664). Both male and female (>6 weeks) mice were used for all experiments, and the number of mice of each sex used to generate each dataset is reported in the legend. The experimenter was blinded to genotype throughout. Genomic DNA was isolated from tail biopsy for genotyping by Transnetyx.

### Immunohistochemistry
#### Substance P and CGRPα staining
Mice were anesthetized with isoflurane and perfused intracardially with heparin then 4% PFA. Tissue was post-fixed in 4% PFA overnight and then cryoprotected in 30% sucrose. Tissue was mounted in OCT and cut using a cryostat into 40–50 µm sections. The sections were incubated in a blocking buffer (5% donkey serum; 0.1% Triton X-100; PBS) for 3 h at room temperature on a shaker. The sections were incubated in 1:500 goat anti-CGRP polyclonal primary antibody (Abcam, #ab36001) or 1:500 rabbit anti substance P polyclonal primary antibody (Abcam, #ab67006) at room temperature overnight. The sections were rinsed twice times with PBS and then incubated for 2 hr in 1:200 Cy5-conjugated donkey anti-rabbit secondary antibody or 1:200 Cy5-conjugated donkey anti-goat secondary antibody (Thermo Fisher Scientific). The sections were rinsed two times with PBS and mounted in ProLong diamond antifade mounting media (Thermo Fisher Scientific) onto slides (Daigger Scientific). Z-stacks were acquired on an Olympus confocal microscope using a 20 x objective and processed using ImageJ/FIJI software (National Institute of Health).

#### Fos staining for neuronal activity
Mice were injected with 10 µl of 0.1% Alexa Fluor 647 conjugated cholera toxin subunit B into one hindpaw so that the most highly-innervated spinal cord sections could later be identified. After 1 week, mice were habituated then the stimulation was performed. For capsaicin-evoked dorsal horn Fos, capsaicin (0.3%) was injected into the right hindpaw of mice in chambers on a plexiglass stand. For Fos elicited by dynamic allodynia stimulation, mice that received a spared nerve injury procedure 3 weeks prior were stimulated with a paint brush while housed on a wire mesh stand (3x10 min stimulation

period, with 1-min rest every 10 min). Note that for SNI experiments, the CTB was injected into the contralateral paw, because severing two branches of the sciatic nerve would have prevented efficient CTB transit in the ipsilateral paw. The mice were perfused as above and spinal cords were harvested 75–120 min after stimulation. The sections were incubated in a blocking buffer (5% goat serum; 0.1% Triton X-100; PBS) for 3 hr at room temperature on a shaker. The sections were incubated in 1:1000 rabbit anti Fos primary antibody (Cell Signaling Technology, Phospho-c-Fos (Ser32) (D82C12) XP Rabbit mAb, #5348) at room temperature overnight. The sections were rinsed two times with PBS and incubated for 2 hr in 1:5000 Alexa Fluor 750 goat anti-rabbit secondary antibody (Life Technologies). The sections were rinsed two times with PBS and mounted in ProLong diamond antifade mounting media (Thermo Fisher Scientific) onto slides (Daigger Scientific). Z-stack images were acquired on an Olympus confocal microscope using a 20 x objective. The five sections with the greatest Alexa Fluor 647 cholera toxin subunit B signal were imaged for each mouse. This ensured Fos activity was measured only in those sections strongly innervated by the hindpaw. The number of Fos-positive nuclei in the dorsal horn were quantified in ImageJ/FIJI software (National Institute of Health) by a blinded observer using a semi-automated procedure. Briefly, brightness and contrast were uniformly adjusted for all images, then the 'Find Maxima' and 'Analyze particles' functions were used within a region of interest to identify Fos puncta. The list of puncta was then manually curated to generate a final estimate of the number of puncat within the region of interest. Fos puncta counts for the five sections were averaged for each mouse.

## Neuropeptide imaging

### Generation and maintenance of Substance P and CGRP-sniffer cell lines

To generate the Substance P-sniffer cell line, we produced a mouse *Tacr1* DNA construct by gene synthesis (Epoch Life Science, GS66243-3). *Tacr1* was subcloned along with a synthesized human G-protein α-subunit gene $G_{\alpha15}$ and *GCaMP6s* it into the lentiviral plasmid backbone pLV-CMV-PGK-Hyg (Cellomics Technology, LVR-1046) to create the final lentiviral plasmid pLV-CMV-GCaMP6s-P2A-TACR1-T2A-hG15-PGK-Hyg. We used this plasmid to produce lentiviral particles (Vigene Biosciences) and infected them into human embryonic kidney cells at a multiplicity of infection of 20 following manufacturer's instructions. The Flp-In T-REx HEK293 cell line was used (Thermo Fisher Scientific, R78007). Stably expressing cells were isolated by treating with 200 μg/mL hygromycin B (Thermo Fisher Scientific, 10687010).

To generate the CGRP-sniffer cell lines, we produced mouse *Calcrl* and *Ramp1* DNA constructs by DNA synthesis (Epoch Life Science). *Calcrl* was subcloned along with a synthesized human G-protein α-subunit gene $G_{\alpha15}$ and GCaMP6s into a dox-inducible backbone (GS66685-1). These were transfected into HEK293 cells using a PiggyBac transposase system and selected for using puromycin. These cells were then transfected with a plasmid pSBbi-Neo-cmv-TETo2-RAMP1 IRES-mTagBFP2-NLS and selected using geneticin. Doxycycline was then used to induce expression of both constructs, and cells were maintained in doxycycline.

The cell lines were maintained on polystyrene culture plates (Thermo Fisher Scientific, 07-200-80) in a 5% $CO_2$ humidified incubator at 37 °C. The growth medium was changed every 2–3 days and consisted of DMEM/F12 (Thermo Fisher Scientific, 11330032) supplemented with 10% fetal bovine serum (Thermo Fisher Scientific, 26140079) and 200 μg/ml hygromycin B. Cells were passaged when they reached confluence, which was roughly twice per week, and were never propagated past 20 passages. For passaging, cells were rinsed in PBS (Thermo Fisher Scientific, 10010023) and then incubated in Accutase (Thermo Fisher Scientific, 00-4555-56) for ~5 min at 37 °C to detach. Cells were collected in a 15 ml tube (Thermo Fisher Scientific, 12-565-268) and centrifuged at 300 rcf for 3 min to pellet. The supernatant was aspirated, and cells were resuspended in growth medium followed by plating in new polystyrene plates. Typical dilution ratios for passaging were between 1:3 and 1:20. For imaging, Substance P-sniffer cells were cultured onto eight-well glass slides, and imaged before they reached confluency.

### Adult dorsal root ganglion culture and Substance P-sniffer co-culture

Dorsal root ganglia (DRG) were dissected from the entire length of the spinal column and then digested in a pre-equilibrated enzyme mix for 35–45 min (37 °C, 5% $CO_2$). The enzyme mix consisted of Hanks' balanced salt solution containing collagenase (type XI; 5 mg/ml), dispase (10 mg/ml), HEPES

(5 mM), and glucose (10 mM). DRGs were then gently centrifuged for 3 min at 300 revolutions per minute, the supernatant was discarded and replaced with warmed DMEM/F-12, supplemented with 10% fetal bovine serum (FBS). Next, DRGs were mechanically triturated with three fire-polished glass Pasteur pipettes of gradually decreasing inner diameter. Dissociated cells were then centrifuged again at 300 revolutions per minute, the supernatant was discarded and cells were re-suspended in the required volume of DMEM supplemented with FBS and nerve growth factor (50 ng/ml). Finally, cells were plated onto 8-well glass slides coated with poly-L-lysine (1 mg/ml) and laminin (1 mg/ml), and incubated at 37 °C in 5% CO2. 24 hr later, Substance P-sniffer cells were resuspended in DMEM/F-12, supplemented with 10% fetal bovine serum (FBS) and nerve growth factor (50 ng/ml). 70,000 cells were dispensed into each imaging well containing DRG neurons, and incubated for at least a further 24 hr before imaging was performed.

## In vitro imaging

Imaging was performed in Ringer's solution: 125 mM NaCl, 3 mM KCl, 5 mM $CaCl_2$, 1 mM $MgCl_2$, 10 mM glucose, and 10 mM HEPES (all from Sigma-Aldrich), adjusted to pH 7.3 with 1 M NaOH, and osmolality measured ~280 mmol/kg. SP-sniffer cells alone, or co-cultured with DRGs, were rinsed in Ringer's solution and imaged in Ringer's solution at room temperature on an Olympus IX73 inverted microscope using a pco.panda sCMOS back-illuminated camera at 2 frames/s. Tac1-RFP was excited at 560 nm LED and GCaMP6f at 488 nm. All imaging trials began with 15 s of baseline measurement and then the cells were treated with chemicals by micropipette. The chemicals used were capsaicin (10 µM) and Substance P amide (1 pM to 1 µM). The peak $\Delta F/F_0$ was calculated to quantify the changes in fluorescence associated with chemical application, with $F_0$ defined as the mean fluorescence intensity of the entire field of view in the 5 s immediately preceding stimulation.

## Behavioral assays

### Von Frey

Punctate mechanical sensitivity was measured using the up-down method of Chaplan to obtain a 50% withdrawal threshold (*Chaplan et al., 1994*). Mice were habituated on a mesh wire stand for 1 hr. A 0.16 g Von Frey hair was applied to the plantar region of the paw for 2 s. A response was recorded when the mouse swiftly lifted its paw in response to the stimulus. A positive response resulted in application of a filament of lesser strength on the following trial, and no response in application of a stronger filament. To calculate the 50% withdrawal threshold, five responses surrounding the 50% threshold were obtained after the first change in response. The pattern of responses was used to calculate the 50% threshold = $(10[\chi + \kappa\delta]/10,000)$, where $\chi$ is the log of the final von Frey filament used, κ=tabular value for the pattern of responses and δ the mean difference between filaments used in log units. The log of the 50% threshold was used to calculate summary and test statistics.

### Pinprick

Mice were habituated on a mesh wire stand for 1 hr. A 27-gauge needle was blunted and used to apply pressure to the hind paw. A withdrawal response was quantified as the mouse lifting the paw swiftly away from the blunted needle. The pinprick stimulation was repeated for 5 or 10 trials with 5-min breaks inbetween. The percentage of withdrawal responses was calculated for each mouse.

### Clip

Mice were habituated in opaque chambers on a plexiglass stand for 1 hr. The mouse was restrained and an alligator clip was applied to the hindpaw just in front of the heel. The mouse was returned to the chamber. The mouse's behavioral response was recorded from below using a video camera. After 60 s, the clip was removed. The amount of time the mouse spent attending the clip and associated paw was quantified post-hoc by a blinded observer. Attending was defined as biting/handling the clip, and biting/licking the paw.

### Hargreaves

Mice were habituated on a plexiglass stand for 1 hr with the Hargreaves machine turned on in a dark room. The radiant heat stimulus was aimed for the centre of the plantar region of the mouse's hind

paw. The active intensity of the low stimulus was 40 intensity units, and of the high stimulus 55. The response latency was recorded for five trials with 10 min breaks in between each trial.

## Hot/cold plate

The plate was set to the desired temperature. A camera set up with a mirror was used for better visibility of the mouse on the hot plate. A clear cylindrical tube was placed around the hot plate to ensure the mouse cannot escape. Once the hot plate reached the set temperature, the mouse was placed on the hot plate and the top of the cylinder was covered. At the cut-off time, the mouse was taken off the hot plate and placed back in the cage. The latency to lick was scored post-hoc using the recorded videos by a blinded observer.

## Dry ice

Mice were habituated on a plexiglass floor stand for 1 hr. A 2 ml plastic syringe was cut in half to allow for crushed dry ice to be compacted into the tube. The dry ice was pushed up against the glass where the mouse's hind paw was resting. Withdrawal latency was recorded with a stopwatch. 10 min were given in between each trial. The procedure was repeated five times and an average of the withdrawal latencies was calculated.

## Acetone

Mice were habituated on a plexiglass floor stand for 1 hr. Mice were videotaped as acetone was applied to the plantar region of their hind paw. Acetone was applied using a custom-made applicator, consisting of a 1 ml syringe and plastic well where acetone was pushed out to generate a droplet for applying to the paw. Behavior was scored post-hoc by a blinded observer and the total amount of time spent licking the paw in a 60 s time interval was recorded. The experiment was repeated for a second trial and average scores were used.

## Chemical algogens

Mice were habituated on a plexiglass stand for 1 hr. 0.3% capsaicin in 80% saline / 10% Tween-80 /10% ethanol or 1% AITC in saline was injected into the plantar surface of the hind paw. The injected volume for both was 20 µl. Mice were video-taped for 5 min. A blinded observer scored the amount of time in seconds the mouse spent licking the injected paw in a 5-min time period.

## Acetic acid

Mice were habituated on a plexiglass stand for 1 hr. 0.6% of acetic acid was injected intraperitoneally at a dose of 10 µl/g weight of the animal. Mice were videotaped for 20 min. The number of writhes was quantified post-hoc by a blinded observer during a 15-min interval between 5 and 20 min after injection.

## Conditioned taste aversion

Mice were single-housed and habituated for 3 days to drink from two glass bottles with stainless-steel ball-bearing spouts. Mice then received 3 days of training, where they were water-deprived for 22 hr followed by 30 min exposure to a single water-containing bottle, then 90 min exposure to both bottles. On the conditioning day, after 22 hr of water deprivation, mice were exposed to a single bottle containing saccharin water (15 mM) for 45 min. Mice were then injected with lithium chloride (200 mg/kg) or PBS, and then given ordinary water for 75 min. The animals were then water deprived for a further 22 hr. On the test day, mice were simultaneously exposed to a bottle containing water and a bottle containing saccharin for 45 min. The volume of each solution consumed was measured by weighing the bottles before and after. To calculate the saccharin preference index of each animal, the volume of saccharin consumed was divided by the total of fluid consumed, with a value less than 0.5 indicating the development of an aversion to saccharin.

## Chloroquine-induced itch

Hair from the nape of the mouse's neck was removed 2–3 days prior to behavioral experiments using VEET hair removal cream and the area thoroughly washed and moisturizing cream applied. Mice were

habituated to transparent acrylic chambers (10cm x 10cm x 13 cm) 30 min prior to injection. Post habituation, a blinded investigator injected 20 µl of 25 µM Chloroquine diphosphate salt (Sigma-Aldrich: C6628) solution in 0.9% saline intradermally into the nape of the neck using a 31 G insulin syringe. Videos were quantified by a separate blinded investigator with a scratching bout counted each time the mouse attended to the injected area with the hind paw after the paw was removed from the floor or mouth.

## Inflammatory pain models

CFA and PGE2 was purchased from Sigma. 20 µl of CFA was injected into the heel of the hind paw. PGE2 (20 µl, 500 nM) was injected into the plantar surface of the hindpaw, and behavior was assessed over the same time-frame that produces in vivo sensitization of nociceptor responses (up to 120 min). Inflammation-induced hypersensitivity was measured using the Hargreaves and the von Frey, as described above.

## Spared nerve injury

The mouse was anesthetized with isofluorane. The mouse was placed under the nose cone, belly down, and the left hind limb area was shaved. The area was wiped clean with an ethanol wipe and coated with betadine. Once the mouse was unresponsive, a 1-inch horizontal cut was made in the skin right along the femur. Once the natural separation of muscles was located, a small scissor tool was used to puncture the fascia and separate out the muscles. Once the sciatic nerve was located, curved forceps were used to separate out the nerve from the muscle. The nerve was traced towards the knee until the branch of three nerves (perineal, tibial, and sural) was found. The sural nerve was spared and the perineal and tibial nerves were cut. The mouse was sutured up and placed back in the homecage for recovery without postoperative analgesics. Mechanical allodynia was assessed using the von Frey assay from 2 to 21 days post surgery.

## Oxaliplatin

Chemotherapy-induced neuropathy was induced in mice by intraplantar injection of oxaliplatin into the left hind paw (**Deuis et al., 2013**; **MacDonald et al., 2021**). Oxaliplatin was made up to a dose of 80 µg in 40 µl of 5% glucose solution, due to its instability in chloride-containing saline solution. The number of nocifensive behaviors in 5 min was assessed on the Cold Plate held at 10 °C 24 hr later.

## Neurogenic inflammation

### Evans blue assay

Mice were anesthetized using isoflurane. 200 µl of Evans Blue dye was injected into the mouse's tail vein. 15 mins later, one hindpaw was injected with an algogen. After a further 30 min, the volume of each paw was measured by displacement using a plethysmometer (Ugo Basile). Both paws were then cut at the ankle and placed in tubes in the oven at 55 °C for 24 hr to dry. 1 ml of formamide was then added to each paw and incubated for 4-6 days at 55 °C to extract the dye from the paw tissue. 50 µl samples of dye-infused formamide from each paw were then dispensed to a 96-well plate in duplicate, along with a range of Evans blue dilutions in formamide to generate a standard curve. The optical density of each sample was then measured using a plate-reader, and the concentration of Evans Blue per paw sample interpolated from the standard curve to generate an index of extravasation in that paw.

## Statistical analysis

A Shapiro-Wilk test was used to test the normality of the data. Data were then compared using two-way ANOVA with post-hoc Sidak's test, Student's unpaired t test or Mann-Whitney U Test. The α value was 0.05. For all experiments $n$ is the number of animals, except Substance P and CGRP imaging where $n$ is the number of wells. Error bars denote standard error of the mean throughout. No power analyses were used to determine sample sizes, but our sample sizes are similar to those from previous studies (**Caterina et al., 2000**; **Lagerström et al., 2010**; **Liu et al., 2010**; **Nassar et al., 2004**). Graphs were generated and statistical analysis was performed using GraphPad 8.0 software

(Prism). All statistical tests and results are reported in **Supplementary file 1**. Source data is deposited on dryad https://doi.org/10.5061/dryad.hqbzkh1tm.

## Acknowledgements
We thank the National Center for Complementary and Integrative Health and the National Institute of Neurological Disorders and Stroke for providing funding. DIM was supported by a European Molecular Biology Organization Postdoctoral Fellowship, a Branco Weiss Fellowship – Society in Science, and an NIH Office of Autoimmune Disease Intramural Award. We are grateful to Nick Ryba and members of the Chesler lab for feedback and advice.

## Additional information

### Competing interests
Alexander Theodore Chesler: Reviewing editor, eLife. The other authors declare that no competing interests exist.

### Funding

| Funder | Grant reference number | Author |
| --- | --- | --- |
| National Center for Complementary and Integrative Health | ZIAAT000028 | Alexander Theodore Chesler |
| Branco Weiss Fellowship - Society in Science | Postdoctoral Fellowship | Donald Iain MacDonald |
| EMBO | Postdoctoral Fellowship | Donald Iain MacDonald |

The funders had no role in study design, data collection and interpretation, or the decision to submit the work for publication.

### Author contributions
Donald Iain MacDonald, Conceptualization, Data curation, Formal analysis, Supervision, Funding acquisition, Investigation, Visualization, Methodology, Writing – original draft, Writing – review and editing; Monessha Jayabalan, Jonathan T Seaman, Rakshita Balaji, Data curation, Investigation, Methodology; Alec R Nickolls, Resources, Data curation, Investigation, Methodology; Alexander Theodore Chesler, Conceptualization, Resources, Data curation, Supervision, Funding acquisition, Investigation, Methodology, Writing – original draft, Project administration, Writing – review and editing

### Author ORCIDs
Donald Iain MacDonald ⓘD http://orcid.org/0000-0002-4863-1093
Alexander Theodore Chesler ⓘD https://orcid.org/0000-0002-3131-0728

### Ethics
This study was performed in strict accordance with the recommendations in the Guide for the Care and Use of Laboratory Animals of the National Institutes of Health. All of the animals were handled according to approved institutional animal care and use committee (IACUC) protocols (#1365; 1369), NINDS.

Reviewer #2 (Public review): https://doi.org/10.7554/eLife.93754.3.sa1
Reviewer #3 (Public review): https://doi.org/10.7554/eLife.93754.3.sa2
Author response https://doi.org/10.7554/eLife.93754.3.sa3

## Additional files

### Supplementary files

Supplementary file 1. Table summarizing all statistical tests used in the manuscript.

MDAR checklist

### Data availability

All data generated or analysed during this study are included in the manuscript and supporting files. Source data files are deposited on dryad https://doi.org/10.5061/dryad.hqbzkh1tm.

The following dataset was generated:

| Author(s) | Year | Dataset title | Dataset URL | Database and Identifier |
|---|---|---|---|---|
| MacDonald D, Chesler AT | 2025 | Data from: Pain persists in mice lacking both Substance P and CGRPα signaling | https://doi.org/10.5061/dryad.hqbzkh1tm | Dryad Digital Repository, 10.5061/dryad.hqbzkh1tm |

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
