## [Editor Report · eLife Assessment]

This report used a new double knockout mouse model to investigate the role of two neuropeptides, substance P and CGRPa, in pain signaling. There is **convincing** evidence that double knockout of these two molecules, both of which have historically been associated with pain, does not affect nociception or acute pain behaviors in males and females. This finding is **fundamental**, as it challenges the hypothesis that these peptides are essential for pain transmission, even when targeted together. This paper will be of interest to those interested in the neurobiology of pain and/or neuropeptide function.

---

## [Referee Report · Reviewer #2 (Public review)]

Summary,

The paper aimed to examine the effect of co-ablating Substance P and CGRPα peptides on pain using Tac1 and Calca double knockout (DKO) mice. The authors observed no significant changes in acute, inflammatory, and neuropathic pain. These results suggest that Substance P and CGRPα peptides do not play a major role in mediating pain in mice. Moreover, they reveal that the lack of behavioral phenotype cannot be explained by the redundancy between the two peptides, which are often co-expressed in the same neuron

Strengths,

The paper uses a straightforward approach to address a significant question in the field. The authors confirm the absence of Substance P and CGRPα peptides at the levels of DRG, spinal cord, and midbrain. Subsequently, they employ a comprehensive battery of behavioral tests to examine pain phenotypes, including acute, inflammatory, and neuropathic pain. Additionally, they evaluate neurogenic inflammation by measuring edema and extravasation, revealing no changes in DKO mice. The data are compelling, and the study's conclusions are well-supported by the results. The manuscript is succinct and well-presented.

---

## [Referee Report · Reviewer #3 (Public review)]

In this study, the authors aimed to determine the role of a global double knockout (DKO) of substance P and CGRPα in modulating acute and chronic pain transmission. After successfully generating and validating the DKO mouse model, they conducted a series of behavioral pain assessments to evaluate the role of these neuropeptides in acute and chronic pain. Despite the well-established involvement of substance P and CGRPα in chronic pain, their findings revealed that the global loss of both neuropeptides did not affect the transmission of either acute or chronic pain.

A major strength of the paper is that they validated their double knockout mouse model before using a comprehensive array of both acute and chronic pain tests to reach their conclusions. One minor weakness is that their n numbers for some of the studies conducted are low.

The conclusions made by the authors are largely supported by their results and the authors successfully achieved their aim of investigating the role of simultaneous inhibition of substance P and CGRPα in pain transmission.

This study offers valuable insights into our understanding of the pain pathways. Both Substance P and CGRPα neuropeptides and their receptors were considered key players in pain signaling due to their high expression in pain-responsive neurons. However, targeting these peptides in clinical trials has not been successful. By investigating the simultaneous inhibition of substance P and CGRPα through the generation of Tac1 and Calca double knockout (DKO) mice, the authors addressed an important gap in the field. Their comprehensive assessment of pain behaviors across a range of acute and chronic pain models revealed an unexpected outcome: the absence of both neuropeptides did not significantly alter pain responses. This finding is pivotal, as it challenges the hypothesis that these peptides are essential for pain transmission, even when targeted together.

Comments on revisions:

All my previous concerns have been addressed.

---

## [Author Response]

The following is the authors’ response to the original reviews.

**Reviewer #1 (Public Review)**:MacDonald et al., investigated the consequence of double knockout of substance P and CGRPα on pain behaviors using a newly created mouse model. The investigators used two methods to confirm knockout of these neuropeptides: traditional immunolabeling and a neat in vitro assay where sensory neurons from either wildtype or double knock are co-cultured with substance P "sniffer cells", HEK cells stably expressing NKR1 (a substance P receptor), GCaMP6s and Gα15. It should be noted that functional assays confirming CGRPα knockout were not performed. Subsequently, the authors assayed double knockout mice (DKO) and wildtype (WT) mice in numerous behavioral assays using different pain models, including acute pain and itch stimuli, intraplanar injection of Complete Freund's Adjuvant, prostaglandin E2, capsaicin, AITC, oxaliplatin, as well as the spared nerve injury model. Surprisingly, the authors found that pain behaviors did not differ between DKO and WT mice in any of the behavioral assays or pain paradigms. Importantly, female and male mice were included in all analyses. These data are important and significant, as both substance P and CGRPα have been implicated in pain signaling, though the magnitude of the effect of a single knockout of either gene has been variable and/or small between studies.The conclusions of the study are largely supported by the data; however, additional experimental controls and analyses would strengthen the authors claims.

We thank the reviewer for their insightful comments and have answered them below.

(1) The authors note that single knockout models of either substance P or CGRPα have produced variable effects on pain behaviors that are study-dependent. Therefore, it would have strengthened the study if the authors included these single knockout strains in a side-by-side analysis (in at least some of the behavioral assays), as has been done in prior studies in the field when using double- or triple-knockout mouse models (for example, see PMID: 33771873). If in the authors hands, single knockouts of either peptide also show no significant differences in pain behaviors, then the finding that double knockouts also do not show significant differences would be less surprising.

In our study, we found no phenotypic differences between WT and DKO mice, suggesting Substance P and CGRPα are largely dispensable for pain behavior. We agree that if we had we observed significant changes in behavior, it would have been interesting to examine the effects of knocking out each gene individually to determine which peptide is responsible for the phenotype. However, given the double deletion had no effect, we can predict that loss of each alone would have no or minor effects. In line with this, a more recent study that comprehensively phenotyped the Calca KO mouse found no deficits in a range of danger related behaviors (PMID: 34376756). Overall, as we are reporting negative data about the Double KO, we do not believe extensive studies of the single KOs is necessary to support the findings of our paper.

(2) It is unclear why the authors only show functional validation of substance P knockout using "sniffer" cells, but not CGRPα. Inclusion of this experiment would have added an additional layer of rigor to the study.

Imaging of CGRPα release is more challenging using the ‘sniffer’ approach because functional CGRP receptors require the expression of two genes: Calcrl (or Calcr) along with Ramp1. We now have succeeded in generating a new stable cell line expressing Calcrl and Ramp1, along with GCaMPs and human Galpha15 and include new data in the revised Figure 1F-H and Figure Supplement 1B. These cells respond robustly to CGRPalpha, but not to SP. In contrast, the existing SP cell line responds to SP but not CGRPalpha. Capsaicin evokes a strong response in these cells in co-culture with DRGs. This response is dramatically reduced in the DKO. This data therefore confirms our mice have a loss of CGRPalpha signaling as indicated by IHC.

(3) The authors should be a bit more reserved in the claims made in the manuscript. The main claim of the study is that "CGRPα and substance P are not required for pain transmission." However, the authors also note that neuropeptides can have opposing effects that may produce a net effect of no change. In my view, the data presented show that double knockout of substance P and CGRPα do not affect somatic pain behaviors, but do not preclude a role for either of these molecules in pain signaling more generally. Indeed, the authors also note that these neuropeptides could be involved in nociceptor crosstalk with the immune or vascular systems to promote headache. The authors only assayed pain responses to glabrous skin stimulation. How the DKO mice would behave in orofacial pain assays, migraine assays, visceral pain assays, or bone/joint pain assays, for example, was not tested. I do not suggest the authors include these experiments, only that they address the limitations/weaknesses of their study more thoroughly.

The reviewer makes an important point that we agree with. Our study assesses acute and chronic pain in peptide DKO mice lacking Substance P and CGRPα. Most of our data focuses on the hindpaw as pain in the paw is the gold-standard approach for phenotyping pain targets and numerous well-validated chronic pain models have been developed for this body site. However, to extend the conclusions to other tissues, we did also look at visceral pain and GI distress using acetic acid and LiCl models (Figure 2J and Figure 2 supplement). We agree with the reviewer that given the utility of CGRP monoclonal antibodies, migraine experiments would be interesting for future studies using these mice, a point we highlight in the discussion. Bone/joint pain is also clearly important from a translational perspective, but outside the scope of the current study.

(4) A more minor but important point, the authors do not describe the nature of the WT animals used. Are the littermates or a separately maintained colony of WT animals? The WT strain background should be included in the methods section.

The WT strain are C57/BL6j from Jackson Lab. This has been added to the methods.

**Reviewer #2 (Public Review):**
Summary:The paper aimed to examine the effect of co-ablating Substance P and CGRPα peptides on pain using Tac1 and Calca double knockout (DKO) mice. The authors observed no significant changes in acute, inflammatory, and neuropathic pain. These results suggest that Substance P and CGRPα peptides do not play a major role in mediating pain in mice. Moreover, they reveal that the lack of behavioral phenotype cannot be explained by the redundancy between the two peptides, which are often co-expressed in the same neuronStrengths:The paper uses a straightforward approach to address a significant question in the field. The authors confirm the absence of Substance P and CGRPα peptides at the levels of DRG, spinal cord, and midbrain. Subsequently, they employ a comprehensive battery of behavioral tests to examine pain phenotypes, including acute, inflammatory, and neuropathic pain. Additionally, they evaluate neurogenic inflammation by measuring edema and extravasation, revealing no changes in DKO mice. The data are compelling, and the study's conclusions are well-supported by the results. The manuscript is succinct and well-presented.

We thank the reviewer for their enthusiasm for the importance of our work.

**Reviewer #3 (Public Review):**
In this study, the authors were assessing the role of double global knockout of substance P and CGPRα on the transmission of acute and chronic pain. The authors first generated the double knockout (DKO) mice and validated their animal model. This is then followed by a series of acute and chronic pain assessments to evaluate if the global DKO of these neuropeptides are important in modulating acute and chronic pain behaviors. Authors found that these DKO mice Substance P and CGRPα are not required for the transmission of acute and chronic pain although both neuropeptides are strongly implicated in chronic pain. This study does provide more insight into the role of these neuropeptides on chronic pain processing, however, more work still needs to be done. (see the comments below).

We thank the reviewer for their detailed and constructive feedback, and below outline the steps we have taken to answer their concerns.

(1) In assessing the double KO (result #1), why are different regions of the brains shown for substance P and CGRPα (for example, midbrain for substance P and amygdala for CGRPα)? Since the authors mentioned that these peptides co-expressed in the brain (as in the introduction), shouldn't the same brain regions be shown for both IHC? It would be ideal if the authors could show both regions (midbrain and amygdala) in addition to the DRG and spinal cord for both peptides in their findings.In addition, since this is double KO, the authors should show more representative IHC-stained brain regions (spanning from the anterior to posterior).

We could not co-stain both SP and CGRP in the same sections as the DKO mouse has endogenous GFP and RFP fluorescence, limiting us to one channel (far red). Specifically, we use a Calca KO that is a Cre:GRP knock-in/knockout (Chen et al 2018, PMID30344042) and Tac1 KO is a tagRFP knock-in/knockout (Wu et al 2018 PMID29485996). This is why we show different brain sections.

(2) It is also unclear as to why the authors only assessed the loss of substance P signaling in the double KO mice. Shouldn't the same be done for CGRPα signaling? Either the authors assess this, or the authors have to provide clear explanations as to why only substance P signaling was assessed.

As noted in our response to Reviewer 1, imaging of CGRP release is more challenging using the ‘sniffer’ approach because functional CGRP receptors require the expression of two genes: Calcrl (or Calcr) along with Ramp1. We have now generated this cell line and performed the experiment (see revised Figure 1 and Figure 1 Supplement).

(3) Has these animal's naturalistic behavior been assessed after the double KO (food intake, sleep, locomotion for example)? I think this is important as changes to these naturalistic behaviors can affect pain processes or outcomes.

We agree that assessment of naturalistic behavior including food intake, sleep and locomotion would be interesting to look at in DKO mice. However, our study is focused on acute and chronic pain behavior of these animals, and therefore a comprehensive phenotypic assessment of naturalistic home-cage behavior is outside the scope of our study.

(4) Figure 2H: The authors acknowledge that there is a trend to decrease with capsaicin-evoked coping-like responses. However, a close look at the graph suggests that the lack of significance could be driven by 1 mouse. Have the authors run an outlier test? Alternatively, the authors should consider adding more n to these experiments to verify their conclusions.

We were reluctant to add more animals searching for significance. Instead, we investigated the potential phenotype further by looking at cfos staining in the cord and found no differences (Figure 2, supplement 1). This result suggests loss of the two peptides does not grossly disrupt capsaicin evoked pain signal transmission between the nociceptor and post-synaptic dorsal neurons in the spinal cord.

(5) Similarly, the values for WT in the evoked cFos activity (Figure 2- Suppl Figure 1) are pretty variable. Considering that the n number is low (n = 5), authors should consider adding more n.Also, since the n number is low in this experiment (eg. 5 vs 4), does this pass the normality test to run a parametric unpaired t-test? Either the authors increase their n numbers or run the appropriate statistical test.

As described in the statistical tables, the Shapiro-Wilk test indicates these data do pass the normality test. Therefore, we retain the use of the unpaired t test, which demonstrates no significant difference between the groups.

(6) In most of the results, authors ran a parametric test despite the low n number. Authors have to ensure that they are carrying out the appropriate statistical test for their dataset and n number.

We now provide a table of the statistical results, which provides detailed information about all statistical tests performed in this study. For experiments where we make a single comparison between the two distributions (WT vs DKO), we have run a Shapiro-Wilk test. Where the data from both groups pass the normality test, we retain the use of the unpaired t test. Where the Shapiro-Wilk test indicates data from either group are unlikely to be normally distributed, we now use a Mann-Whitney U test to compare the groups, as this non-parametric test makes no assumptions about the underlying distribution.

Many experiments involved two factors (genotype, and e.g. temperature, drug, time-point). These data were analyzed in the original submission using 2-WAY ANOVA or Repeated Measures 2-WAY ANOVA, followed by post-hoc Sidak’s tests to compute p values adjusted for multiple comparisons. Because there is no widely agreed non-parametric alternative to 2-WAY ANOVA for analyzing data with two factors and that enables us to account for multiple comparisons, we used 2-WAY ANOVA as is typically used in the field for these kinds of experiments. We reasoned sticking with the 2-WAY ANOVA was the best course of action based on information provided by the statistical software used for this study - https://www.graphpad.com/support/faq/with-two-way-anova-why-doesnt-prism-offer-a-nonparametric-alternative-test-for-normality-test-for-homogeneity-of-variances-test-for-outliers/

We note that regardless of the test, our conclusion that there are no major changes in acute or chronic pain behaviors are clear and strongly supported.

(7) Along the same line of comment with the previous, authors should increase the n number for DKO for staining (Figure 4) as n number is only 3 and there is variability in the cFos quantification in the ipsilateral side.

We believe this is not necessary as the finding is clear that there is no difference.

(8) Authors should provide references for statement made in Line 319-321 as authors mentioned that there are accumulating evidence indicating that secretion of these neuropeptides from nociceptor peripheral terminals modulates immune cells and the vasculature in diverse tissues.

We now provide several references to primary papers and reviews supporting this statement.

(9) Authors state that the sample size used was similar to those from previous studies, but no references were provided. Also, even though the sample sizes used were similar, I believe that the right statistic test should be used to analyze the data.

We have now cited several classic studies phenotyping mouse KOs in pain in the methods that used similar sample sizes. As detailed above, we have taken the reviewer’s feedback on board and performed normality testing to ensure the correct statistical test is used for each experiment.

(10) In the discussion, the authors noted that knocking out of a gene remains the strongest test of whether the molecule is essential for a biological phenomenon. At the same time, it was acknowledged that Substance P infusion into the spinal cord elicits pain, but it is analgesic in the brain. The authors might want to expand more on this discussion, including how we can selectively assess the role of these neuropeptides in areas of interest. For example, knocking out both Substance P and CGRPα in selected areas instead of the global KO since there are reported compensatory effects.

This is highlighted in the closing paragraph: “Emerging approaches to image and manipulate these molecules (Girven et al., 2022; Kim et al., 2023), as well as advances in quantitating pain behaviors (Bohic et al., 2023; MacDonald and Chesler, 2023), may ultimately reveal the fundamental roles of neuropeptides in generating our experience of pain.” The Kim preprint (now published, and so the citation has been updated in the text) describes a method of inactivating neuropeptide transmission in select brain regions in a cell-type specific manner.

**Recommendations for the authors:**

**Reviewer #2 (Recommendations For The Authors):**
I do not have any major comments. My minor comments are as follows:(1) What was the control group for all behavioral studies? Was it WT from an independent colony or one of the littermates was used for generating controls?

We used C57/Bl6 mice from Jax. This is now mentioned in methods.

(2) In Fig. 2H, it seems that the effect will become significant if several mice are added.

We are reluctant to add mice searching for significance. Sample sizes were determined before we collected the data blind.

(3) There is no figure 3, but two figures 4.

Thank you. This has been corrected.

(4) Multiple typos in the legend for figure 4 (lines 234-254). Line 242 (& n=8 (3M, 3F)), line 243 (swelling and plasma), line 252 ((n=8 for) & n=6 for DKO (4M, 4F)).

Thank you. This has been corrected.

(5) In Figure 4 (lines 273-285), the contralateral side is mentioned in B but no images are shown.

Thank you. We removed the mention.

(6) Although ligand knockouts cannot be compared directly with receptor inhibition, the readers could benefit from discussing studies of receptor ablation and/or pharmacological inhibition.

We do discuss the classic studies of receptor KO, and the clinical data on receptor blockers here –

“However, selective antagonists of the Substance P receptor NKR1 failed to relieve chronic pain in human clinical trials (Hill, 2000). Although CGRP monoclonal antibodies and receptor blockers have proven effective for subsets of migraine patients, their usefulness for other types of pain in humans is unclear (De Matteis et al., 2020; Jin et al., 2018). In line with this, knockout mice deficient in Substance P, CGRPα or their receptors have been reported to display some pain deficits, but the analgesic effects are neither large nor consistent between studies (Cao et al., 1998; De Felipe et al., 1998; Guo et al., 2012; Salmon et al., 2001, 1999; Zimmer et al., 1998).”

**Reviewer #3 (Recommendations For The Authors):**
Minor comments:(1) Figure 1E: What does chambers mean? Additionally, are the 12 chambers equally from the male and female samples (6 from male and 6 from female)?

We have changed this to well. Each replicate is an individual well from 8 well chamber slide. In all these experiments, the wells are approximately evenly distributed by mouse, because from each mouse we cultured around 8 wells’ worth of DRGs.

(2) Figure 1D: What does low and high mean in the Hargreaves test?

These refer to a low and high active intensity of the radiant heat stimulus. Number is now described in the methods. 40 and 55 in the intensity units used by the instrument.

(3) Figure 2-Suppl Figure 1: Authors should provide a bigger image of the image so that it is clearer to the readers.

We think the image is of a reasonable size and comparable to the images used elsewhere in the paper.

(4) Authors should consider labeling their supplementary figures in running numbers or combining supplementary figures together to avoid confusion. For example, Figure 2-Supplementary Figure 1 and Figure 2- Supplementary Figure 2 can be combined as just Supplementary Figure 2.

We agree with the reviewer this would be clearer, but we have followed eLife’s convention for labelling and numbering supplements.

(5) Figure 3 is mislabeled as Figure 4.

Thank you. We have corrected this.

(6) Only female mice were used in the CFA experiment, which does not go in line with the rest of the results which consist of both sexes.

We have repeated the experiment with additional male mice. To be consistent with the von frey data, these were followed for 7 days, and so the figure now shows a 7 day time course.

(7) Typo in line 243. The word "and" is subscript.

Thank you. We have corrected this.

(8) There is a typo in the legend for Figure 4 where E is labeled I, G is labeled as F, and J is labeled as J.

Thank you. We have corrected this.

(9) Authors should specify what "several weeks" means (Line 263).

It means three weeks. We tested to 21 days. We will replace with three.

(10) Authors should specify what "one day" means (Line 267). For example, how many days after the intraplantar oxaliplatin treatment? Also, authors should justify why that specific time point was selected or have a reference for it.

This means one day after - 24 hours. Please see PMID: 33693512. Two references are provided in them methods.

(11) Figure 4 legend: authors should again be specific on what "prolonged" entails (Line 277).

We have replaced prolonged with 30 minutes brushing. Specifically, 3 x 10 min stim period, with 1 min rest between stim. It is in the methods.

(12) In the methods section, authors state that both male and female mice were used for all experiments. However, only female mice were used in the CFA experiment (see minor comment #6). Authors should verify and correct this.

This is correct. We only used female mice for one of the groups. We have since repeated with males, now included in the data.

(13) Authors should be more specific in the methods section on how long the habituation per day, how many days and what were the mice habituation to (experimenter, room, chamber, etc)?

As noted in the methods, mice are habituated for at least an hour to the chambers, and thus implicitly to the room. We do not perform explicit habituation to the investigator such as repeated handling.

(14) Authors need to provide more information on the semi-automated procedure they are referring to in Line 397. Also, authors should also provide the criteria for cFos quantification (eg. Intensity, etc). If this has been published before, they should provide the reference.

We have added this. We used the ‘Find maxima’ and ‘Analyze particles’ functions in FIJI, followed by a manual curation step.

(15) How much acetone was applied and how was it applied to the paw? (Line 495)

We used the same applicator (1ml syringe with a well at the top) to generate a droplet of acetone that was used for all mice. This has been added to methods.

(16) Authors should specify the amount of capsaicin injected in μl (Line 500).

20 ul. We have added this.

(17) Authors should explain or reference why they are analyzing the 15 min interval between 5 and 20 minutes for injection (Line507-508).

Acetic acid behaviour lasts around 30 mins in our hands. We chose the 15 minute interval because it reduces burdensome hand scoring time by 50% versus doing the whole 30 mins. We reasoned that in the first 5 mins post injection the animal behaviour may be contaminated by stress related to handling, injection and return to chamber. Thus, 5 and 20 minutes provided a sensible time-frame for scoring the behavior when it is at its peak.

(18) Authors have to provide more information/explanation on how they decide on the conditioned taste aversion protocol. Like why they do 30 mins exposure to a single water-containing bottle followed 90 mins exposure to both bottles. If this has been published before, they should provide the reference.

We read dozens of different published protocols in the literature, and piloted one that was something of an amalgam of some of them with various adaptations of convenience. Because it worked on our first attempt, we stuck to it. The advantage of the CTA assay is it is incredibly robust to changes in the specificities of the paradigm, evincing the clear survival value of learning to avoid tastes that make you sick.

(19) Authors again should provide more detail in their methods section.a. Specify the time frame that they are assessing here (Line 533).

This can be seen in the Figure. 0 to 120 mins. We have added it to the methods.

b. How long were the mice allowed to recover post-SNI before mechanical allodynia was assessed (Line 545)?

This is apparent in the figures. 2 days to 21 days. We have added it to the methods.

c. How much of the oxaliplatin was injected into the mice?

40 ug / 40 ul (see PMID:33693512)

Editors note: Reviewers agreed that addressing the concerns about power, outliers, and statistics, as well as functional validation of CGRPα would raise the strength of evidence to compelling, and inclusion of comparison to single KO would raise it to exceptional.Should you choose to revise your manuscript, please check to ensure full statistical reporting including exact p-values wherever possible alongside the summary statistics (test statistic and df) and 95% confidence intervals. These should be reported for all key questions and not only when the p-value is less than 0.05.